# Remote Non-Invasive Fabry-Pérot Cavity Spectroscopy for Label-Free Sensing

**DOI:** 10.3390/s23010385

**Published:** 2022-12-29

**Authors:** Abeer Al Ghamdi, Benjamin Dawson, Gin Jose, Almut Beige

**Affiliations:** 1School of Physics and Astronomy, University of Leeds, Leeds LS2 9JT, UK; 2School of Physics and Astronomy, King Saud University, Riyadh 11362, Saudi Arabia; 3School of Chemical and Process Engineering, University of Leeds, Leeds LS2 9JT, UK

**Keywords:** cavity sensing, optical sensors, photonic sensors

## Abstract

One way of optically monitoring molecule concentrations is to utilise the high sensitivity of the transmission and reflection rates of Fabry-Pérot cavities to changes of their optical properties. Up to now, intrinsic and extrinsic Fabry-Pérot cavity sensors have been considered with analytes either being placed inside the resonator or coupled to evanescent fields on the outside. Here we demonstrate that Fabry-Pérot cavities can also be used to monitor molecule concentrations non-invasively and remotely, since the reflection of light from the target molecules back into the Fabry-Pérot cavity adds upwards peaks to the minima of its overall reflection rate. Detecting the amplitude of these peaks reveals information about molecule concentrations. By using an array of optical cavities, a wide range of frequencies can be probed at once and a unique optical fingerprint can be obtained.

## 1. Introduction

Sensors which offer high specificity and reliably monitor substances in chemical, physical and biological systems play a vital role in many applications. For example, biosensors (e.g., Refs. [1,2,3,4,5,6]) are used in tissue and cell analysis, microbiological investigations and drug improvement studies. To answer the ever-growing demand for highly sensitive and selective measurement devices which do not require large lab-based equipment, Fabry-Pérot cavity sensors have already attracted considerable attention [7,8]. These sensors can be divided into two main categories: *intrinsic* and *extrinsic* Fabry-Pérot cavity sensors. In the case of intrinsic sensors [9,10,11], which are the most common, the sample to be measured is placed inside the resonator (Figure 1a). In the case of extrinsic sensors [12,13,14,15,16,17,18,19], the sample changes the optical properties of the resonator when in contact with one of its mirrors on the outside (Figure 1b).

Fabry-Pérot cavities are optical resonators with two highly-reflecting mirrors separated by a gap of length $L_{0}$. When monochromatic light enters the cavity, it bounces back and forth between the mirrors many times before eventually leaking out. The amount of light that passes through the resonator depends on the frequency of the incoming light compared to the distance of the mirrors. More concretely, the incoming laser light accumulates a phase factor during each round trip. If the cavity is in resonance, this phase factor is an integer multiple of 2π. Hence, all light travelling in the forward direction interferes constructively, while light travelling in the backwards direction interferes destructively, such that all light eventually leaves the cavity on the opposite side [20]. Moreover, in the case of mirrors with absorption and with reduced reflection rates, the transmission peak is broadened, and a wider range of frequencies is transmitted.

The characteristic reflection and transmission spectrum of a Fabry-Pérot cavity can be characterised for example by the quality factor *Q* with $Q=\omega_{\mathrm{cav}}/\Delta \omega$, where $\omega_{\mathrm{cav}}$ denotes the cavity resonance frequency and Δω characterises the line width. In general, the higher the *Q* factor of a cavity, the more sensitive are $\omega_{\mathrm{cav}}$ and Δω to the presence of molecule concentrations. For example, some Fabry-Pérot cavity sensors take advantage of the effect of the presence of analytes with distinctive optical transitions on mirror reflection rates or of refractive index changes inside the resonator. In general, information about concentrations can be deduced by comparing frequency shifts and the broadening of the line width of the transmitted light signal to a known baseline [8].

Unfortunately, intrinsic Fabry-Pérot cavity sensors tend to suffer from low coupling efficiency. In the presence of analytes, the sensor is likely to require a re-alignment of its mirrors in order to produce reliable results, which is experimentally demanding [9,10,11]. Extrinsic Fabry-Pérot cavity sensors overcome this problem by never changing the inside of the resonator and have already been used to perform highly sensitive measurements of small molecules such as proteins and DNA [12,13,14,15,16,17,18]. However, like intrinsic Fabry-Pérot cavity sensors, they are invasive and still need to be in close contact with the sample to make a measurement, which strongly limits their applications.

To overcome this problem, this paper proposes a Fabry-Pérot cavity sensor, which can perform *remote* measurements of molecule concentrations. As we shall see below, the sensor can be used to continuously and non-invasively monitor a wide range of physical, chemical and biological processes. No direct contact with the target molecules is required as long as they are optically accessible. Hence, remote Fabry-Pérot cavity sensors might even be suitable to continuously and non-invasively monitor biomolecule concentrations [14,15,21,22]. Alternatively, they could be used to monitor the contents of transparent containers and bottles without the need to open them, which has applications, for example, in the food industry.

The basic design of the remote Fabry-Pérot cavity sensor, which we propose in this paper, is shown in Figure 2. In the following, we require that the target molecules have optical transitions and therefore reflect some of the transmitted laser light back into the resonator. The environment of the analytes needs to be transparent—or at least semitransparent—in the relevant frequency range. In contrast to extrinsic and intrinsic Fabry-Pérot cavity sensors, remote Fabry-Pérot cavity sensors explore the high sensitivity of their reflection rate R(ω) to the presence of stray light to measure target molecule concentrations. This is possible, since the interference inside the cavity changes when the analytes reflect some of the already transmitted light back into the resonator.

Suppose the Fabry-Pérot cavity is ideal and the incoming laser light is on resonance with the resonator. In this case, all light is eventually transmitted and the reflection rate R(ω)=0. Hence, the only effect that the stray light coming from the target molecules can have is to increase R(ω). In general, if the reflecting molecules are randomly distributed within the sample and cover an area wider than the wavelength of the reflected light, the reflected light accumulates equally distributed random phases, and the resulting R(ω) depends only on the optical properties of the cavity mirrors and on the concentration and the optical properties of the reflecting atomic particles.

As we shall see below, randomly distributed molecules therefore add narrow upwards peaks to minima of the reflection spectrum of the Fabry-Pérot cavity. These peaks can be detected and their height reveals information about molecular concentrations. The higher the concentration of the target molecules, the larger the amplitude of the upwards peaks that are added to the minima of the measurement signal. As one might expect, the amplitude of these peaks is exactly the same as the amplitude of the reflected light in the absence of the Fabry-Pérot cavity. A main purpose of the resonator is to filter out one specific frequency component of the reflected light.

However, notice also that the reflection rate of the target molecules depends strongly on their atomic level structure. Simultaneously probing the response of a remote Fabry-Pérot cavity sensor to different frequencies of light can therefore provide a unique optical fingerprint, which increases the selectivity of the sensor. This is an important feature of remote Fabry-Pérot cavity sensors, since cavity resonance frequencies can be varied much more easily than the frequency of particular receptor molecules. For example, this can be conducted by varying the length of the cavity or by changing the angle of the incident light. Moreover, as illustrated in Figure 3, a single sensor could contain an array of optical cavities with different resonance frequencies. Another important advantage of remote Fabry-Pérot cavity sensors is that they do not need to be adjustable, since they do not need to be stabilised in direct contact with the target molecules. This means that they can be fabricated more easily, for example by integrating them into optical fibers [23,24,25,26], while accompanying them with integrated mode-matching optics [27].

This paper contains five sections. Section 2 reviews the optical properties of Fabry-Pérot cavities. Afterwards, in Section 3, we calculate the overall reflection rates of mirror arrays, which contain at least three mirrors with the help of a scattering matrix approach [28,29]. In Section 4, we use the results obtained in Section 3 to predict the overall reflection rate of the proposed remote Fabry-Pérot cavity sensor in Figure 2 and study the dependence of this rate on molecule concentrations. Finally, we summarise our findings in Section 5.

## 2. The Reflection Rates of Fabry-Pérot Cavities

To be more realistic in our predictions, we consider in the following asymmetric mirrors with coherent light absorption and allow the media on both sides of an interface to have a different refractive index. For simplicity, we only consider light propagating along the *x*-axis. To introduce the notation for studying light scattering by Fabry-Pérot cavities, we first have a closer look at the case of a single mirror. As illustrated in Figure 4, we assume that the mirror is in contact with air on one side and with a dielectric medium with a refractive index n≠1 on the other.

### 2.1. The Electromagnetic Field in Air and in a Dielectric Medium

A change in refractive index alters the electric field amplitudes and the frequencies of incoming wave packets. However, in this subsection, we show that it is possible to simply ignore this effect as long as we are only interested in overall reflection rates. The reason for this is that we can always replace an experimental setup, which contains a medium, with one that only contains air. The predicted interference effects are the same in both cases as long as the dimensions of the system are changed accordingly. Not having to pay attention to refractive index changes simplifies the analysis in the following sections, but we will need to keep in mind that all (complex) reflection and transmission rates in this paper refer to the case where mirrors are placed in the air.

As usual, we describe the dynamics of the electromagnetic field in a dielectric medium with permittivity ε and permeability μ in the absence of any charges and currents by Maxwell’s equations. For light propagating along the *x* axis, these predict that
(1)$$\frac{\partial^{2}}{\partial x^{2}}E\left(x,t\right)=\varepsilon \mu \frac{\partial^{2}}{\partial t^{2}}E\left(x,t\right),$$
where E(x,t) denotes the electric field amplitude at position *x* and time *t*. From classical electrodynamics, we know that the basic solutions of Maxwell’s equations are plane travelling waves which can be superposed to form wave packets of any shape that travel at the speed of light *c* [30],
(2)$$c={\left(\varepsilon \mu \right)}^{-1/2}.$$

Suppose s=±1 indicates the respective direction of propagation, λ=1,2 denotes the polarisation and *k* is a positive wave number. Then, the basic solutions of Maxwell’s equations for the electric field amplitudes $E_{sk\lambda }\left(x,t\right)$ for given parameters (s,k,λ) can be written as
(3)$$E_{sk\lambda }\left(x,t\right)=E_{0}e^{ik(x-sct)}+c.c.$$

In the following, we refer to $E_{0}$ as the complex electric field amplitude for left- and for right-moving light. In the case of air, we have $\varepsilon =\varepsilon_{0}$ and $\mu =\mu_{0}$. However, in a general dielectric medium, the refractive index *n*,
(4)$$n={\left(\varepsilon \mu /\varepsilon_{0}\mu_{0}\right)}^{1/2},$$
is different from n=1. In general, the electric field E(x,t) are superpositions of the above electric field amplitudes $E_{sk\lambda }\left(x,t\right)$.

For light propagating along the *x*-axis, we therefore obtain the equivalence relation [31]
(5)$$E_{\mathrm{med}}\left(x,t\right)={\left(n^{3}\varepsilon_{0}/\varepsilon \right)}^{1/2}E_{\mathrm{air}}\left(nx,t\right).$$

Using Equation (Equation 1), it is relatively easy to check that if $E_{\mathrm{air}}\left(x,t\right)$ solves Maxwell’s equations in air, then $E_{\mathrm{med}}\left(x,t\right)$ in Equation (Equation 5) solves Maxwell’s equations in a dielectric medium and vice versa. Moreover, suppose $E_{\mathrm{med}}\left(x,t\right)$ describes the electric field in a medium of length *L* and with an area *A* around the *x*-axis, while $E_{\mathrm{air}}\left(x,t\right)$ describes the electric field in a volume of air of length *nL* and with an area $n^{2}A$ around the *x* axis. Then, one can show, using Equation (Equation 5), that
(6)$$A\int_{0}^{L}dx\varepsilon E_{\mathrm{med}}{\left(x,t\right)}^{2}=A\int_{0}^{L}dLn^{3}\varepsilon_{0}E_{\mathrm{air}}{\left(nx,t\right)}^{2}=n^{2}A\int_{0}^{nL}dx\varepsilon_{0}E_{\mathrm{air}}{\left(x,t\right)}^{2}.$$

This means the factor on the right-hand side of Equation (Equation 5) has been chosen, such that the electric field energy is exactly the same in both cases.

### 2.2. The Reflection and Transmission Rates of a Single Mirror

The observations in the previous subsection allows us to model the effect of the two-sided semitransparent mirror in Figure 4 by simply replacing it with an analogous mirror placed in air. In the following, we denote the (complex) transmission and reflection rates of this effective mirror by $t_{a}^{\left(1\right)}$, $t_{b}^{\left(1\right)}$, $r_{a}^{\left(1\right)}$ and $r_{b}^{\left(1\right)}$, respectively. The superscript ^(1)^ helps to distinguish these rates from the rates of other setups with more than one mirror present. The reflection and transmission rates for light approaching the mirror from different directions are in general not the same. As we shall see below, they differ in general by a phase factor. In the presence of absorption, they differ also in size.

In the following, we describe light scattering by a semi-transparent mirror, and later also by a Fabry-Pérot cavity and other mirror arrays, by time-independent scattering operators. Suppose $E_{i}^{\mathrm{in}}\left(x,t\right)$ and $E_{i}^{\mathrm{out}}\left(x,t\right)$ with i=a,b are the complex electric field amplitudes of the incoming and of the outgoing laser light on both sides of an interface, which has been placed at x=0. In the case of *real* transmission and reflection rates, $E_{a}^{\mathrm{out}}\left(x,t\right)$ and $E_{a}^{\mathrm{out}}\left(x,t\right)$ are given by
(7)$$\begin{array}{ccc} E_{a}^{\mathrm{out}}\left(x,t\right) & = & r_{a}^{\left(1\right)}E_{a}^{\mathrm{in}}\left(-x,t\right)+t_{b}^{\left(1\right)}E_{b}^{\mathrm{in}}\left(x,t\right),\\ E_{b}^{\mathrm{out}}\left(x,t\right) & = & t_{a}^{\left(1\right)}E_{a}^{\mathrm{in}}\left(x,t\right)+r_{b}^{\left(1\right)}E_{b}^{\mathrm{in}}\left(-x,t\right). \end{array}$$

However, here we need to consider *complex* rates. Their phase factors are later chosen such that energy is always conserved. The reason for these phases is that the complex electric field amplitude $E_{0}$ in Equation (Equation 3) is defined with respect to x=0 and t=0, while the mirrors are placed at x=0 and x=L, respectively. In the case of *complex* transmission and reflection rates, by definition,
(8)$$\begin{array}{ccc} E_{a}^{\mathrm{out}}e^{ik(x+ct)}+c.c. & = & r_{a}^{\left(1\right)}E_{a}^{\mathrm{in}}e^{ik(-x-ct)}+t_{b}^{\left(1\right)}E_{b}^{\mathrm{in}}e^{ik(x+ct)}+c.c.,\\ E_{b}^{\mathrm{out}}e^{ik(x-ct)}+c.c. & = & t_{a}^{\left(1\right)}E_{a}^{\mathrm{in}}e^{ik(x-ct)}+r_{b}^{\left(1\right)}E_{b}^{\mathrm{in}}e^{ik(-x+ct)}+c.c. \end{array}$$

This applies for all times *t* when
(9)$$(\begin{array}{c} E_{a}^{\mathrm{out}}\\ E_{b}^{\mathrm{out}*} \end{array})=(\begin{array}{cc} r_{a}^{(1)*} & t_{b}^{\left(1\right)}\\ t_{a}^{(1)*} & r_{b}^{\left(1\right)} \end{array})(\begin{array}{c} E_{a}^{\mathrm{in}*}\\ E_{b}^{\mathrm{in}} \end{array}).$$

The energy of the incoming and of the outgoing light are only the same when
(10)$$\begin{array}{ccc} |E_{a}^{\mathrm{out}}{|}^{2}+{\left|E_{b}^{\mathrm{out}}\right|}^{2} & = & |E_{a}^{\mathrm{in}}{|}^{2}+{\left|E_{b}^{\mathrm{in}}\right|}^{2}. \end{array}$$

This equation only holds for all possible electric field amplitudes when
(11)$$\begin{array}{ccc} |r_{a}^{\left(1\right)}{|}^{2}+|t_{a}^{\left(1\right)}{|}^{2}=|r_{b}^{\left(1\right)}{|}^{2}+|t_{b}^{\left(1\right)}{|}^{2} & = & 1,\\ t_{b}^{\left(1\right)}r_{a}^{\left(1\right)}+t_{a}^{\left(1\right)}r_{b}^{\left(1\right)} & = & 0. \end{array}$$

Hence, the phases of the above scattering matrix elements need to be chosen carefully for energy to be conserved. For example, we could assume that
(12)$$r_{b}^{\left(1\right)}=-r_{a}^{\left(1\right)},t_{b}^{\left(1\right)}=t_{a}^{\left(1\right)}.$$

In the presence of absorption, the energy of the outgoing light must be smaller than the energy of the incoming light. In this case, the equal sign in Equation (Equation 10) is replaced by a smaller-equal sign and the reflection and transmission rates can assume a wider range of values [32,33,34]. Mapping the above outgoing electric field amplitudes onto the corresponding amplitudes in a medium can be conducted with the help of Equation (Equation 3).

### 2.3. Fabry-Pérot Cavities

Next, we consider two parallel semitransparent mirrors, $M_{1}$ and $M_{2}$, separated by a distance $L_{0}$. As illustrated in Figure 5, we denote the reflection and transmission rates of these mirrors in air by $r_{i}^{\left(2\right)}$ and $t_{i}^{\left(2\right)}$, respectively, with i=a,b,c,d. To determine the effect of both mirrors, we now need to take into account that the complex electric field amplitudes $E_{i}^{\mathrm{in}}$ with i=a,d of monochromatic light accumulate phase factors $e^{\pm i\phi_{0}}$ with
(13)$$\phi_{0}=n_{0}L_{0}k=n_{0}L_{0}\omega /c_{0}$$

When travelling the length of the cavity. Here, $n_{0}=1$ and *k* and ω denote the wave number and the frequency of the incoming light and $c_{0}$ is the speed of light in air. Which sign applies depends on the respective direction of travel.

Suppose light approaches the Fabry-Pérot cavity only from the right (i.e., $E_{a}^{\mathrm{in}}=0$) and $E_{a}^{\mathrm{out}}$ and $E_{d}^{\mathrm{out}}$ are the complex electric field amplitudes of outgoing light. Then, by definition, we now have
(14)$$\begin{array}{ccc} E_{d}^{\mathrm{out}}e^{ik(x-ct)}+c.c. & = & r_{d}^{\left(2\right)}E_{d}^{\mathrm{in}}e^{ik(2L_{0}-x+ct)}+c.c.\\ & & +t_{c}^{\left(2\right)}t_{d}^{\left(2\right)}r_{b}^{\left(2\right)}\sum_{N=0}^{\infty }{\left(r_{b}^{\left(2\right)}r_{c}^{\left(2\right)}\right)}^{N}E_{d}^{\mathrm{in}}e^{ik(-x-2NL_{0}+ct)}+c.c., \end{array}$$

If the two mirrors of the Fabry-Pérot cavity are placed at x=0 and at $x=L_{0}$. This equation holds at all times *t*, when
(15)$$E_{d}^{\mathrm{out}*}=\left[r_{d}^{\left(2\right)}e^{2i\phi_{0}}+t_{c}^{\left(2\right)}t_{d}^{\left(2\right)}r_{b}^{\left(2\right)}\sum_{N=0}^{\infty }{\left(r_{b}^{\left(2\right)}r_{c}^{\left(2\right)}e^{-2i\phi_{0}}\right)}^{N}\right]E_{d}^{\mathrm{in}}.$$

Having a look also at other cases, we therefore find that
(16)$$\left(\begin{array}{c} E_{a}^{\mathrm{out}}\\ E_{d}^{\mathrm{out}*} \end{array})=(\begin{array}{cc} S_{11}^{\left(2\right)} & S_{12}^{\left(2\right)}\\ S_{21}^{\left(2\right)} & S_{22}^{\left(2\right)} \end{array})(\begin{array}{c} E_{a}^{\mathrm{in}*}\\ E_{d}^{\mathrm{in}} \end{array}\right)$$
with the scattering matrix elements
(17)$$\begin{array}{ccc} S_{11}^{\left(2\right)} & = & r_{a}^{(2)*}+t_{a}^{(2)*}t_{b}^{(2)*}r_{c}^{(2)*}e^{-2i\phi_{0}}\sum_{N=0}^{\infty }{\left(r_{b}^{(2)*}r_{c}^{(2)*}e^{-2i\phi_{0}}\right)}^{N},\\ S_{12}^{\left(2\right)} & = & t_{b}^{\left(2\right)}t_{d}^{\left(2\right)}\sum_{N=0}^{\infty }{\left(r_{b}^{\left(2\right)}r_{c}^{\left(2\right)}e^{-2i\phi_{0}}\right)}^{N}, \end{array}$$
while
(18)$$\begin{array}{ccc} S_{21}^{\left(2\right)} & = & t_{a}^{(2)*}t_{c}^{(2)*}\sum_{N=0}^{\infty }{\left(r_{b}^{(2)*}r_{c}^{(2)*}e^{-2i\phi_{0}}\right)}^{N},\\ S_{22}^{\left(2\right)} & = & r_{d}^{\left(2\right)}e^{2i\phi_{0}}+t_{c}^{\left(2\right)}t_{d}^{\left(2\right)}r_{b}^{\left(2\right)}\sum_{N=0}^{\infty }{\left(r_{b}^{\left(2\right)}r_{c}^{\left(2\right)}e^{-2i\phi_{0}}\right)}^{N}. \end{array}$$

Since the factors $r_{i}^{\left(2\right)}r_{j}^{\left(2\right)}$ in this equation are in general smaller than one, the above sums can be simplified using the geometric series equation. Doing so, we find that
(19)$$\begin{array}{c} S_{11}^{\left(2\right)}=r_{a}^{(2)*}+\frac{t_{a}^{(2)*}t_{b}^{(2)*}r_{c}^{(2)*}e^{-2i\phi_{0}}}{1-r_{b}^{(2)*}r_{c}^{(2)*}e^{-2i\phi_{0}}},S_{12}^{\left(2\right)}=\frac{t_{b}^{\left(2\right)}t_{d}^{\left(2\right)}}{1-r_{b}^{\left(2\right)}r_{c}^{\left(2\right)}e^{-2i\phi_{0}}},\\ S_{21}^{\left(2\right)}=\frac{t_{a}^{(2)*}t_{c}^{(2)*}}{1-r_{b}^{(2)*}r_{c}^{(2)*}e^{-2i\phi_{0}}},S_{22}^{\left(2\right)}=r_{d}^{\left(2\right)}e^{2i\phi_{0}}+\frac{t_{c}^{\left(2\right)}t_{d}^{\left(2\right)}r_{b}^{\left(2\right)}}{1-r_{b}^{\left(2\right)}r_{c}^{\left(2\right)}e^{-2i\phi_{0}}}. \end{array}$$

In the absence of absorption and gain, the energy of the incoming and of the outgoing light must be the same. In analogy to Equation (Equation 10), this now applies when
(20)$$|E_{a}^{\mathrm{out}}{|}^{2}+|E_{d}^{\mathrm{out}}{|}^{2}=|E_{a}^{\mathrm{in}}{|}^{2}+{\left|E_{d}^{\mathrm{in}}\right|}^{2}.$$

Substituting Equation (Equation 16) into this equation, we observe that the scattering matrix elements of the Fabry-Pérot cavity must be such that
(21)$$\begin{array}{ccc} |S_{11}^{\left(2\right)}{|}^{2}+|S_{21}^{\left(2\right)}{|}^{2}=|S_{12}^{\left(2\right)}{|}^{2}+|S_{22}^{\left(2\right)}{|}^{2} & = & 1,\\ S_{11}^{(2)*}S_{12}^{\left(2\right)}+S_{21}^{(2)*}S_{22}^{\left(2\right)} & = & 0, \end{array}$$

In analogy to Equation (Equation 11). Because of Equation (Equation 19), these conditions now only hold when all transmission and reflection rates are real, when $r_{i}^{(2)2}$ and $t_{i}^{(2)2}$ add up to one for all *i* and when
(22)$$t_{a}^{\left(2\right)}=t_{b}^{\left(2\right)},t_{c}^{\left(2\right)}=t_{d}^{\left(2\right)},r_{b}^{\left(2\right)}=-r_{a}^{\left(2\right)},r_{d}^{\left(2\right)}=-r_{c}^{\left(2\right)}.$$

Suppose laser light enters the Fabry-Pérot cavity in Figure 5 only from the left and $E_{d}^{\mathrm{in}}=0$. Using the above equations, one can then demonstrate that the overall reflection rate $R^{\left(2\right)}\left(\omega \right)$ in the absence of absorption is simply given by [35]
(23)$$R^{\left(2\right)}\left(\omega \right)={\left|E_{a}^{\mathrm{out}}\right|}^{2}/{\left|E_{a}^{\mathrm{in}}\right|}^{2}={\left|S_{11}^{\left(2\right)}\right|}^{2}={\left|\frac{r_{a}^{\left(2\right)}+r_{c}^{\left(2\right)}e^{2i\phi_{0}}}{1+r_{a}^{\left(2\right)}r_{c}^{\left(2\right)}e^{2i\phi_{0}}}\right|}^{2}.$$

When the distance $L_{0}$ tends to zero, the cavity mirrors turn into a single mirror interface and $\phi_{0}=0$. For symmetry reasons and since we do not want $R^{\left(2\right)}\left(\omega \right)$ to assume a minimum in a case when there is essentially only a single mirror, the reflection rates $r_{a}^{\left(2\right)}$ and $r_{c}^{\left(2\right)}$ need to have the same sign. Again, in the presence of absorption, a wider range of mirror parameters can be taken into account.

Figure 6 shows the overall reflection rate $R^{\left(2\right)}\left(\omega \right)$ in Equation (Equation 23) as a function of ω for different symmetric mirrors without absorption. When $r_{a}^{\left(2\right)}$ and $r_{c}^{\left(2\right)}$ are the same, $R^{\left(2\right)}\left(\omega \right)=0$ when $\cos(2\phi_{0})=-1$. This applies, for example, when $\phi_{0}$ equals π and the length $L_{0}$ of the cavity equals half the wavelength of the incoming light. In general, these minima occur at angles $\phi_{0}$, which are exactly 2π apart. As expected, the spectral response is sharply peaked about the cavity resonance frequencies when the mirror reflections rates $|r_{a}^{\left(2\right)}|$ and $|r_{c}^{\left(2\right)}|$ are close to one. Lower reflections rates increase the line width of the reflection spectrum. Absorption, moreover, decreases the amplitude of the reflection peak, but the general shape of the curves remains the same.

## 3. The Overall Reflection Rates of Different Mirror Arrays

In this Section, we study the effect of additional mirrors on the reflection rate $R^{\left(2\right)}\left(\omega \right)$ of the Fabry-Pérot cavity in Figure 5. In the following, we are especially interested in the case where a relatively large collection of tiny, randomly-distributed mirrors is placed behind the resonator.

### 3.1. The Overall Reflection Rates of Three-Mirror Systems

However, first we have a closer look at the three-mirror system with mirrors $M_{1}$, $M_{2}$ and $M_{3}$ in Figure 7. To analyse their optical response, we first replace the mirrors $M_{1}$ and $M_{2}$ by a single effective mirror and denote the reflection and transmission rates of this effective mirror by $r_{i}^{\left(3\right)}$ and $t_{i}^{\left(3\right)}$ with i=a,b. (The superscript ^(3)^ indicates that these rates describe an effective mirror in a three-mirror setup). Suppose the mirrors $M_{1}$ and $M_{2}$ have the same optical properties as the mirrors $M_{1}$ and $M_{2}$ of the Fabry-Perot cavity in Figure 5. Then, we can use the results that we obtained in Section 2.3 to show that
(24)$$r_{a}^{\left(3\right)}=S_{11}^{(2)*},t_{a}^{\left(3\right)}=S_{21}^{(2)*},r_{b}^{\left(3\right)}=S_{22}^{\left(2\right)},t_{b}^{\left(3\right)}=S_{12}^{\left(2\right)}$$
with $S_{11}^{\left(2\right)}$, $S_{12}^{\left(2\right)}$, $S_{21}^{\left(2\right)}$ and $S_{22}^{\left(2\right)}$ given in Equation (Equation 16). Moreover, we denote the reflection rate of $M_{3}$ in the following by $r_{c}^{\left(3\right)}$. As in the previous section, *k* and ω denote the wave number and the frequency of the incoming laser light.

Next we notice that the effective mirror and the additional mirror $M_{3}$ form a Fabry-Pérot cavity of length $L_{1}$ which contains a medium with a refractive index $n_{1}\neq 1$ (Figure 7). To simplify our discussion, we further replace this Fabry-Pérot cavity in the following by a Fabry-Pérot cavity of length $n_{1}L_{1}$, which contains air (cf. discussion in Section 2.1). Now suppose $E_{i}^{\mathrm{in}}$ and $E_{i}^{\mathrm{out}}$ with i=a,d denote the incoming and outgoing (complex) electric field amplitudes near the respective mirror surface in air and the phase $\phi_{1}$ is given by
(25)$$\phi_{1}=n_{1}L_{1}k=n_{1}L_{1}\omega /c_{0}.$$

Combining Equations (Equation 19) and (Equation 24), one can now show that
(26)$$\left(\begin{array}{c} E_{a}^{\mathrm{out}}\\ E_{d}^{\mathrm{out}} \end{array})=(\begin{array}{cc} S_{11}^{\left(3\right)} & S_{12}^{\left(3\right)}\\ S_{21}^{\left(3\right)} & S_{22}^{\left(3\right)} \end{array})(\begin{array}{c} E_{a}^{\mathrm{in}}\\ E_{d}^{\mathrm{in}} \end{array}\right)$$

With the scattering matrix element $S_{11}^{\left(3\right)}$ given by
(27)$$S_{11}^{\left(3\right)}=S_{11}^{\left(2\right)}+\frac{S_{12}^{(2)*}S_{21}^{\left(2\right)}r_{c}^{(3)*}e^{-2i\phi_{1}}}{1-S_{22}^{(2)*}r_{c}^{(3)*}e^{-2i\phi_{1}}}.$$

When the setup is only driven by monochromatic laser light from the left, the reflection rate $R^{\left(3\right)}\left(\omega \right)$ of the three-mirror interferometer therefore equals
(28)$$R^{\left(3\right)}\left(\omega \right)=|S_{11}^{\left(3\right)}{|}^{2}={\left|S_{11}^{(2)*}+\frac{S_{12}^{\left(2\right)}S_{21}^{(2)*}r_{c}^{\left(3\right)}e^{2i\phi_{1}}}{1-S_{22}^{\left(2\right)}r_{c}^{\left(3\right)}e^{2i\phi_{1}}}\right|}^{2}.$$

As long as the reflection rate of the third mirror is relatively small, we expect that this rate resembles the reflection rate $R^{\left(2\right)}\left(\omega \right)$ of a Fabry-Pérot cavity relatively well. How much it changes in the presence of the third mirror $M_{3}$ depends, for example, on the frequency of the incoming light and on the distance $L_{1}$ between $M_{2}$ and $M_{3}$.

### 3.2. The Effect of a Randomly Positioned Third Mirror

In this paper, we are especially interested in the case where the distance $L_{1}$ of the third mirror varies randomly over a range that is much larger than the wavelength of the incoming laser light. In this case, the corresponding reflection rate $\bar{R^{\left(3\right)}\left(\omega \right)}$ of the three-mirror system in Figure 7 depends no longer on $L_{1}$. It can be obtained by averaging over all possible values of $\phi_{1}$,
(29)$$\bar{R^{\left(3\right)}\left(\omega \right)}=\frac{1}{2\pi }\int_{0}^{2\pi }d\phi_{1}R^{\left(3\right)}\left(\omega \right).$$

A closer look at Equation (Equation 28) shows that we do not need to know the phase of the reflection rates $r_{c}^{\left(3\right)}$ of the additional mirror, since the dependence on this phase disappears when the above average is taken.

In the absence of absorption, the mirror parameters of the Fabry-Pérot cavity need to be in agreement with Equations (Equation 21) and (Equation 22). These imply for example that $S_{12}^{\left(2\right)}=S_{21}^{\left(2\right)}$. Equation (Equation 28) can therefore be used to show that
(30)$$R^{\left(3\right)}\left(\omega \right)={\left|S_{11}^{(2)*}+|S_{12}^{\left(2\right)}{|}^{2}r_{c}^{\left(3\right)}e^{2i\phi_{1}}\sum_{n=0}^{\infty }{\left(S_{22}^{\left(2\right)}r_{c}^{\left(3\right)}e^{2i\phi_{1}}\right)}^{n}\right|}^{2}.$$

Hence performing the integration in Equation (Equation 29), we obtain the average reflection rate
(31)$$\begin{array}{ccc} \bar{R^{\left(3\right)}\left(\omega \right)} & = & |S_{11}^{\left(2\right)}{|}^{2}+|S_{12}^{\left(2\right)}{|}^{4}\cdot |r_{c}^{\left(3\right)}{|}^{2}\sum_{n=0}^{\infty }{\left|S_{22}^{\left(2\right)}r_{c}^{\left(3\right)}\right|}^{2n}\\ & = & |S_{11}^{\left(2\right)}{|}^{2}+\frac{|S_{12}^{\left(2\right)}{|}^{4}}{1-|S_{22}^{\left(2\right)}r_{c}^{\left(3\right)}{|}^{2}}|r_{c}^{\left(3\right)}{|}^{2}. \end{array}$$

From Equations (Equation 21) and (Equation 22), we also observed that $|S_{12}^{\left(2\right)}{|}^{2}=1-{\left|S_{11}^{\left(2\right)}\right|}^{2}$ and $|S_{11}^{\left(2\right)}{|}^{2}={\left|S_{22}^{\left(2\right)}\right|}^{2}$. These relations can be used to simplify Equation (Equation 31) into
(32)$$\bar{R^{\left(3\right)}\left(\omega \right)}-R^{\left(2\right)}\left(\omega \right)=\frac{{\left[1-R^{\left(2\right)}\left(\omega \right)\right]}^{2}}{1-R^{\left(2\right)}\left(\omega \right)|r_{c}^{\left(3\right)}{|}^{2}}|r_{c}^{\left(3\right)}{|}^{2}.$$

For relatively small values of $r_{c}^{\left(3\right)}$, the overall reflection rates $R^{\left(2\right)}\left(\omega \right)$ and $\bar{R^{\left(3\right)}\left(\omega \right)}$ given in Equations (Equation 23) and (Equation 32) are essentially the same. Moreover, it is relatively straightforward to check that $\bar{R^{\left(3\right)}\left(\omega \right)}=|r_{c}^{\left(3\right)}{|}^{2}$ at the resonance frequency of the cavity, where all incoming laser light is normally transmitted and $R^{\left(2\right)}\left(\omega \right)=0$.

Figure 8 shows the difference $\bar{R^{\left(3\right)}\left(\omega \right)}-R^{\left(2\right)}\left(\omega \right)$ as a function of the frequency ω for different values of $|r_{c}^{\left(3\right)}|$ and illustrates clearly that the presence of a third randomly-positioned mirror adds small upwards peaks to the usual minima of the reflection spectrum of the Fabry-Pérot cavity. This is not surprising, since, in the absence of absorption, $R^{\left(2\right)}\left(\omega \right)$ is zero at the cavity resonance frequencies. Hence, anything that changes the amount of interference between the mirrors can only have one effect, namely an increase in the overall reflection rate of the system. As one would expect, these peaks increase in size but also become slightly broader as $|r_{c}^{\left(3\right)}|$ increases.

### 3.3. The Effect of a Relatively Large Number of Tiny, Randomly-Positioned Mirrors

In this final subsection, we consider the experimental setup in Figure 9 which contains a single Fabry-Pérot cavity as well as a relatively large collection of *N* tiny, identical, randomly-positioned, weakly-reflecting mirrors. In analogy to the previous subsection, we denote the reflection rate of the additional mirrors by $r_{c}^{\left(3\right)}$. Moreover, ΔA and ΔL denote the surface area of a single tiny mirror and the optical depth of the sample. In addition, α denotes the area covered by the incoming laser field. Hence, V=α·ΔL is the size of the total illuminated and randomly occupied volume.

Before calculating the reflection rate R(ω) of the experimental setup in Figure 9, we first consider the case with only one randomly positioned tiny mirror present in *V*. In this case, the probability $P_{1}\left(1\right)$ of finding this mirror within a small volume ΔV=ΔA·ΔL equals
(33)$$\begin{array}{ccc} P_{1}\left(1\right) & = & \frac{\Delta A}{\alpha }. \end{array}$$

Hence, the probability $P_{1}\left(0\right)$ for this ΔV *not* to contain the mirror is
(34)$$P_{1}\left(0\right)=1-\frac{\Delta A}{\alpha }.$$

Now, suppose there are *N* identical tiny reflectors in the volume *V*. In this case, the probability $P_{N}\left(0\right)$ for *not* finding a tiny mirror in a given volume ΔV becomes
(35)$$P_{N}\left(0\right)={\left(1-\frac{\Delta A}{\alpha }\right)}^{N}.$$

Hence, the overall reflection rate R(ω) of the experimental setup in Figure 9 is given by
(36)$$R\left(\omega \right)=P_{N}\left(0\right)R^{\left(2\right)}\left(\omega \right)+\left(1-P_{N}\left(0\right)\right)\bar{R^{\left(3\right)}\left(\omega \right)}.$$

This applies since $P_{N}\left(0\right)$ is also the probability for a tiny laser beam with cross section ΔA
*not* to encounter a tiny mirror. In this case, the reflection rate equals $R^{\left(2\right)}\left(\omega \right)$. Moreover, $1-P_{N}\left(0\right)$ is the probability that the thin laser beam meets a randomly positioned tiny mirror and that its reflection rate equals $\bar{R^{\left(3\right)}\left(\omega \right)}$.

Next, we notice that $P_{N}\left(\geq 1\right)=1-P_{N}\left(0\right)$ is the probability for *at least one* mirror present in a given volume ΔV. This observation allows us to write Equation (Equation 36) as
(37)$$R\left(\omega \right)-R^{\left(2\right)}\left(\omega \right)=P_{N}\left(\geq 1\right)\left[\bar{R^{\left(3\right)}\left(\omega \right)}-R^{\left(2\right)}\left(\omega \right)\right].$$

An analytical expression for $\bar{R^{\left(3\right)}\left(\omega \right)}-R^{\left(2\right)}\left(\omega \right)$ can be found in Equation (Equation 31). For very small reflection rates $r_{c}^{\left(3\right)}$, the reflection rates R(ω) and $R^{\left(2\right)}\left(\omega \right)$ are essentially the same. However, as illustrated in Figure 10, the reflection rate no longer becomes zero at the cavity resonance frequencies. Instead, the minimum of the curves are now given by
(38)$$R_{\min}=P_{N}\left(\geq 1\right)|r_{c}^{\left(3\right)}{|}^{2}.$$

This rate increases, as the reflectivity and the number *N* of the tiny mirrors increases until saturation sets in and $P_{N}\left(\geq 1\right)$ tends to one. Notice also that $R_{\min}$ is the reflection rate that the tiny, randomly-positioned mirrors would have in the absence of the Fabry-Pérot cavity. The main purpose of the resonator is to filter one frequency. As we shall see below, we can deduce additional information about the optical properties of the mirrors from the shape of R(ω) in the presence of the resonator.

## 4. Remote Fabry-Pérot Cavity Spectroscopy

In this final section, we utilise the above-described interference effects for label-free sensing. Figure 11 shows an alternative schematic view of the remote Fabry-Pérot cavity sensor in Figure 2. Similar to the experimental setup in Figure 9, the sensor contains a laser-driven Fabry-Pérot cavity. Its transmitted light approaches the target molecules, which can be located some distance away from the sensor from the left. Our measurement signal is the overall reflection rate R(ω) of the device. A comparison of the experimental setups in Figure 9 and Figure 11 shows that both have the same optical response when:
The target molecules closely resemble tiny, semitransparent mirrors, which reflect at least some of the incoming light back into the Fabry-Pérot cavity without changing its frequency. This applies to a very good approximation, if the frequency of the laser falls within their resonance fluorescence spectrum. As we have observed above, it does not matter whether the reflected light accumulates a random phase in the reflection process. It anyway accumulates a random phase due to the randomness of the position of every molecule within the sample.Moreover, the environment surrounding the target molecules should be mostly transparent to the incoming light. If the environment reflects some of the incoming light even in the absence of the target molecules, the sensor needs to be more sensitive and needs to be more carefully calibrated before measurements can be performed.The target molecules are randomly distributed within the finite volume *V*, as it applies, for example, naturally when they are dissolved in a liquid.

Under these conditions, we can use the reflection rate R(ω), which we derived in the previous section to obtain the overall reflection rate R(ω) of the remote Fabry-Pérot cavity sensor in Figure 11 in the presence of analytes. All we need to do is to replace the variables *N*, ΔA and $r_{c}^{\left(3\right)}$ by the total number of target molecules in the illuminated sample, the average scattering cross section of a single target molecule and its reflection rate, respectively.

### 4.1. Optical Signatures of the Presence of Target Molecules

Section 3 suggests that the reflection of light from the target molecules back into the resonator adds small upwards peaks to the minima of the overall reflection rate R(ω) of the remote Fabry-Pérot cavity sensor. This can be detected, since the amplitude of the minima is no longer zero but equals instead $R_{\min}$ in Equation (Equation 38). To produce visible peaks, the frequency of the driving laser light must lie within the resonance fluorescence spectrum of the target molecules. If the incoming laser light is *not* in resonance, the target molecules become transparent and their reflection rate $r_{c}^{\left(3\right)}$ becomes zero. The specificity of remote Fabry-Pérot cavity sensors comes from the fact that the reflection rate $r_{c}^{\left(3\right)}$ depends strongly on the atomic level structure of the target molecules.

As we can observe from Figure 8 and Figure 10, the amplitude of these additional peaks is in general relatively small. In addition to measuring $R_{\min}$, we therefore recommend to plot the difference $R\left(\omega \right)-R^{\left(2\right)}\left(\omega \right)$ between the measured signal R(ω) and the overall reflection rate $R^{\left(2\right)}\left(\omega \right)$ of the Fabry-Pérot cavity in the absence of target molecule concentrations. This difference has a distinct shape. From Equations (Equation 32) and (Equation 37), we observe that the so-called *full width at half the maximum* (FWHM) of the upwards peak near a cavity resonance frequency depends only on the molecule reflection rate $|r_{c}^{\left(3\right)}{|}^{2}$. It is therefore possible to deduce $|r_{c}^{\left(3\right)}{|}^{2}$ from the measurement signal and to obtain additional information about the species of the analytes.

How much light of a given frequency and polarisation is reflected by the target molecule depends on the strength of their dipole moments and on the level spacings of their energy eigenstates. Every molecule has its own unique resonance fluorescence spectrum and therefore also its own frequency-dependent reflection rate $r_{c}^{\left(3\right)}=r_{c}^{\left(3\right)}\left(\omega \right)$. This observation can be exploited to further enhance the specificity of remote Fabry-Pérot cavity sensors. For example, a sensor which simultaneously probes the response of a sample to several laser frequencies should be enough to distinguish any species with atomic transitions in the optical regime. One way of implementing this idea is to incorporate an array of cavities into the sensor design, as illustrated in Figure 3.

### 4.2. The Dependence of Reflection Rates on Molecule Concentrations

Given the above definitions of the variables *N*, *L* and α, the number density of the target molecules equals
(39)$$C=\frac{N}{V}=\frac{N}{\alpha \cdot \Delta L}$$
when the particles are placed in air. (As discussed in Section 2.1, a correction is needed, if the particles are hosted in a medium with a refractive index n that is different from one.) Hence the probability $P_{N}\left(\geq 1\right)$ in Equation (Equation 37), which now coincides with the relative amount of laser light that encounters at least one target molecule within the illuminated sample, depending in general on *C*. By measuring how the overall reflection rate R(ω) of a remote Fabry-Pérot cavity sensor changes at the resonance frequency, it is therefore also possible to obtain information about target molecule concentrations. As we can observe from Equation (Equation 38), this can be conducted by measuring the minimum reflection rate $R_{\min}$ of the sensor.

For example, suppose every volume element ΔV=ΔA·ΔL contains in general at most one target molecule. This applies to a very good approximation if the scattering cross section ΔA of a single molecule multiplied by the total number of molecules *N* within the sample is much smaller than the laser cross section α, i.e.,
(40)N·ΔA≪α.

In this case, the probability $P_{N}\left(0\right)$ in Equation (Equation 35) simplifies to $P_{N}\left(0\right)=1-N\Delta A/\alpha$. Hence, $P_{N}\left(\geq 1\right)=N\Delta A/\alpha$ to a very good approximation and Equation (Equation 37) becomes
(41)$$\begin{array}{ccc} R\left(\omega \right)-R^{\left(2\right)}\left(\omega \right) & = & \left[\bar{R^{\left(3\right)}\left(\omega \right)}-R^{\left(2\right)}\left(\omega \right)\right]\frac{N\Delta A}{\alpha }\\ & = & \left[\bar{R^{\left(3\right)}\left(\omega \right)}-R^{\left(2\right)}\left(\omega \right)\right]\Delta A\cdot \Delta L\cdot C \end{array}$$

Up to first order in *C*. Since $\bar{R^{\left(3\right)}\left(\omega \right)}-R^{\left(2\right)}\left(\omega \right)$ in Equation (Equation 32) does not depend on *C*, this difference depends linearly on the molecular concentrations.

If we want to measure *C* with accuracy, the concentration of the target molecules should therefore not be too high. Ideally, *C* should be such that a relatively high percentage of the analytes sees the incoming laser light. If the concentration *C* becomes too high, all of the incoming laser field is reflected by molecules and $P_{N}\left(\geq 1\right)=1$. In this case, the sensor saturates and the overall reflection rate R(ω) depends no longer on *C*. In general, remote Fabry-Pérot cavity sensors need to be calibrated carefully, since $R_{\min}$ in Equation (Equation 38) depends also on the molecular scattering cross section ΔA and the the optical depth ΔL of the sample.

## 5. Conclusions

This paper takes advantage of the fact that Fabry-Pérot cavities are very sensitive to any changes that affect the interference of light inside the resonator. However, in contrast to intensive and extensive Fabry-Pérot cavity sensors, we do not rely on refractive index changes or on changes of the reflection and transmission rates of its mirrors. Instead, our main research hypothesis is that randomly distributed atomic particles diffract laser light in a similar fashion as tiny, randomly distributed semitransparent mirrors. The remote Fabry-Pérot cavity sensor works since its overall reflection rate R(ω) changes in a unique way when any of the outgoing light is reflected back into the cavity. Here, the only difference between semitransparent mirrors and the target molecules is that reflection rates by the latter is in general weaker and depends more strongly on the frequency and possibly also on the polarisation of the incoming light.

In the absence of any target molecules, the reflection rate R(ω) of a Fabry-Pérot cavity sensor assumes a minimum at the cavity resonance frequency. For example, in the case of an ideal cavity, all incoming resonant light is transmitted and R(ω)=0. In the presence of the target molecules, this minimum is reduced and a small upwards peak is added, which can be detected. The new minimum reflection rate $R_{\min}$ depends on the size of the reflection rate $r_{c}^{\left(3\right)}$ and the concentration *C* of the target molecules. The strong dependence of $|r_{c}^{\left(3\right)}|$ on the resonance fluorescence spectrum of the target molecules contributes to the selectivity of the proposed sensing device. Moreover, the size of the upwards peak provides information about molecular concentrations.

Although this is a theoretical paper, we expect that the proposed remote Fabry-Pérot cavity sensor can be used for the non-invasive, label-free detection of molecule concentrations in a wide range of scenarios. For example, as we have observed above, the sample that contains the target molecules does not need to be in direct contact with the resonator. Moreover, by probing a wide range of frequencies with an array of optical cavities, a unique optical fingerprint of the target molecules can be obtained. However, some conditions need to hold for remote Fabry-Pérot cavity sensors to work:Optical access to the sample that contains the molecules is required.The laser frequency and therefore also the resonance frequency of the incoming light should lie within the resonance frequency spectrum of the molecules, such that they absorb and re-emit light at that frequency with a relatively high rate.The concentrations of the molecules should be neither too low nor too high to obtain a significant response without saturating the device.The positions of the target molecules should be sufficiently random in order to remove any dependence of the sensor reflection rate R(ω) on the exact distances between the Fabry-Pérot cavity and the molecules.

Since these requirements can be met, at least in principle, we are optimistic that the idea which we present here will find a wide range of applications in sensing physical, chemical, and biological processes in real time, remotely and without perturbing them. Being limited mainly by their quality factor, remote Fabry-Pérot cavity sensors might even find applications as difficult as monitoring molecule concentrations in the human blood [14,15,21,22], if they can be engineered and calibrated well enough to work in realistic uncertain environments. 

## Figures and Tables

**Figure 1 sensors-23-00385-f001:**
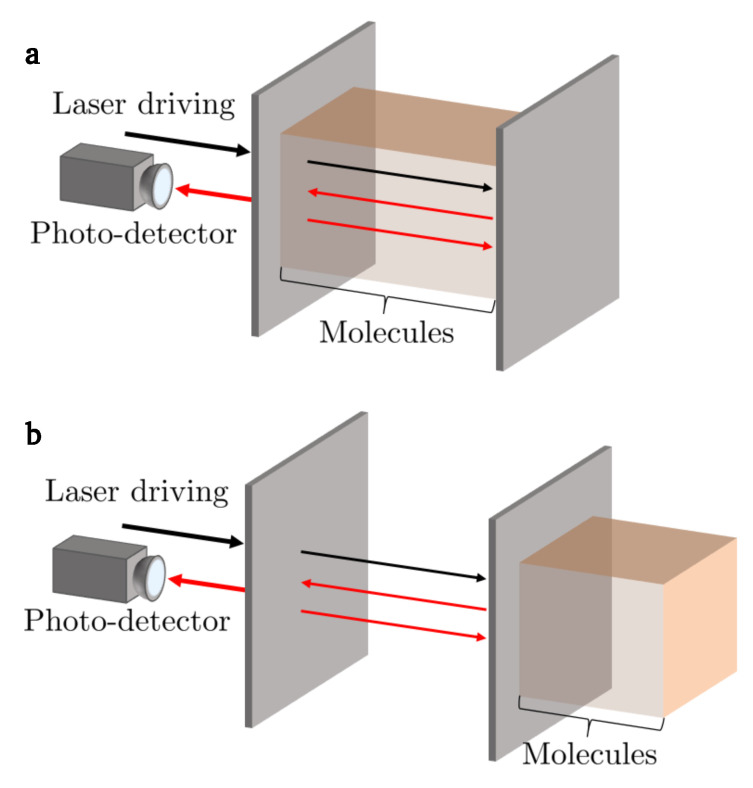
Current Fabry-Pérot cavity sensors can be divided into two main categories. (**a**) In the case of intrinsic sensors, the target molecules are placed on the inside, where they alter the effective cavity length via a change in refractive index. When driven by a laser field, the resonance frequency of the cavity shifts and the line width of the signal can broaden. (**b**) In the case of extrinsic sensors, the target molecules are placed on the outside of one of the cavity mirrors in order to alter its reflection rate, thereby also changing the optical properties of the cavity.

**Figure 2 sensors-23-00385-f002:**
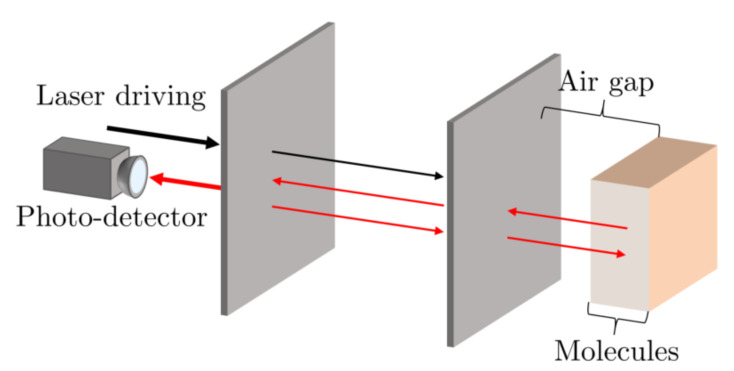
In this paper, we propose an alternative way of using Fabry-Pérot cavities in sensing applications. Here, the substance which we want to analyse is some distance away from the resonator. Hence, we refer to this type of sensor in the following as a remote Fabry-Pérot cavity sensor. If the sample contains atomic particles with optical transitions near the resonance frequency of the sensor, the coherent back reflection of light affects the overall transmission rate of the system. The above experimental setup effectively consists of many mirrors. The measurement signal is an effective frequency-dependent reflection rate of the molecules and, as we shall see below, provides information about their concentration.

**Figure 3 sensors-23-00385-f003:**
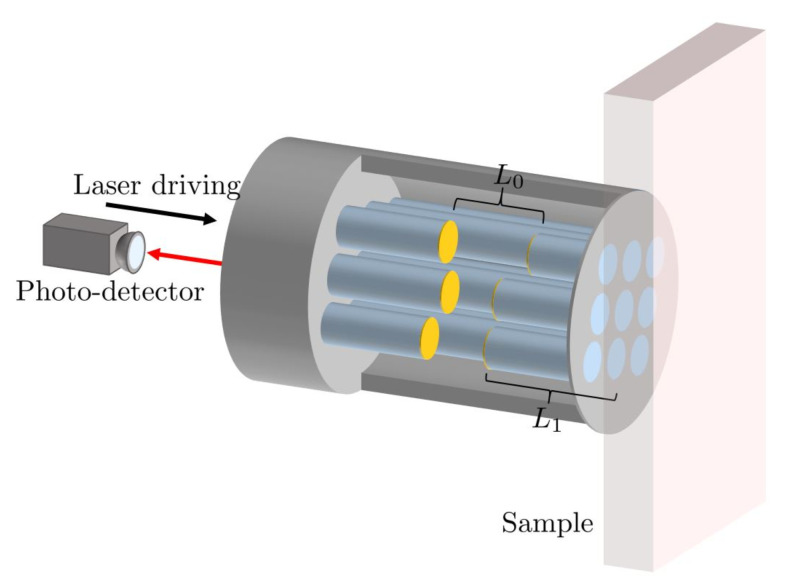
Alternative design of a remote sensor with increased selectivity, which allows for different resonant frequencies to be probed at once. The sensor contains a bundle of optical fibres encased in a protective sheath and embedded with cavities of different lengths $L_{0}$. By measuring the resultant change for each resonance frequency from a known baseline, the type and concentration of the sample to be measured can be determined more easily.

**Figure 4 sensors-23-00385-f004:**
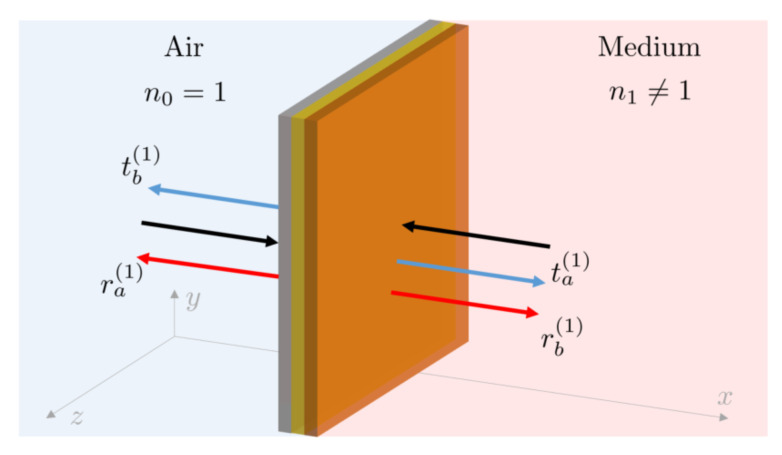
Schematic view of an asymmetric mirror interface with coherent light absorption. The mirror consists of a reflecting layer which may be covered on both sides by thin layers of absorbing material. On the right-hand side, it is attached to a dielectric medium with a refractive index $n_{1}\neq 1$. On the left, the mirror interface borders on air with a refractive index $n_{0}=1$. Since light approaching the mirror from different sides might experience different absorption rates, the overall reflection and transmission rates of the mirror interface, $r_{a}^{\left(1\right)}$, $r_{b}^{\left(1\right)}$, $t_{a}^{\left(1\right)}$ and $t_{b}^{\left(1\right)}$, are in general not the same, even when referring to the case with the mirror being placed in air.

**Figure 5 sensors-23-00385-f005:**
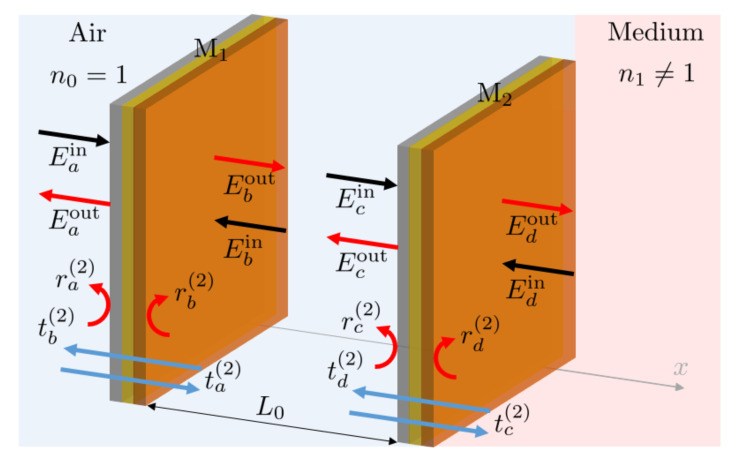
Schematic view of a Fabry-Pérot cavity, which consists of two mirrors, M1 and M2, with a distance $L_{0}$ between them. On its right-hand side, the cavity borders with a medium with a refractive index $n_{1}\neq 1$. All other spaces are filled with air. As in Figure 4, the $r_{i}^{\left(2\right)}$ and $t_{i}^{\left(2\right)}$ denote transmission and reflection rates, while the $E_{i}^{\mathrm{in}}$ and $E_{i}^{\mathrm{out}}$ with i=a,b,c,d denote complex electric field amplitudes.

**Figure 6 sensors-23-00385-f006:**
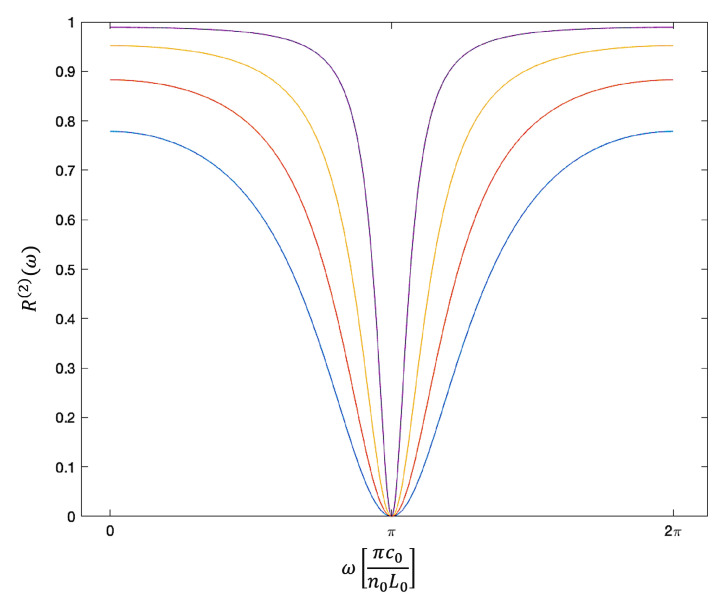
The dependence of the overall reflection rate $R^{\left(2\right)}\left(\omega \right)$ in Equation (Equation 23) on the on the frequency ω of the incoming light. Here, we consider symmetric mirrors without absorption and choose all reflection and transmission rates as suggested in Equation (Equation 22). In addition, we assume that $|r_{a}^{\left(2\right)}{|}^{2}={\left|r_{c}^{\left(2\right)}\right|}^{2}=0.36$ (blue), 0.49 (red), 0.64 (yellow) and 0.81 (purple). The figure shows the typical reflection spectrum of a Fabry-Pérot cavity. At the resonance frequency of the resonator, $R^{\left(2\right)}\left(\omega \right)=0$, independent of the reflection rates of the two mirrors. Increasing the mirror reflection rates, increases the quality factor *Q* of the cavity and results in a narrower downwards peak in the cavity resonance fluorescence spectrum R(ω).

**Figure 7 sensors-23-00385-f007:**
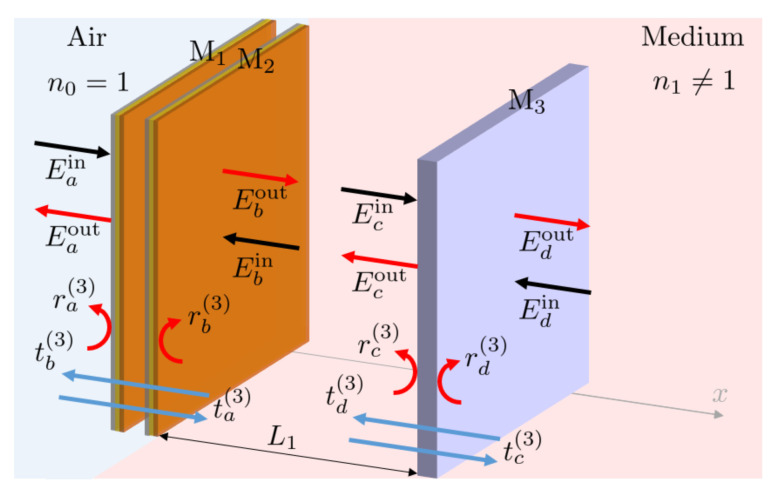
Schematic view of a three-mirror system, which contains two mirrors $M_{1}$ and $M_{2}$, separated by a distance of $L_{0}$ and a third mirror, $M_{3}$, a distance of $L_{1}$ away from the $M_{1}$–$M_{2}$ cavity. In a realistic scenario, the setup might be attached to a dielectric medium with a refractive index n≠1. The figure also shows the relevant electric field amplitudes $E_{i}^{\mathrm{in}}$ and $E_{i}^{\mathrm{out}}$ with i=a,b,c,d near the relevant mirror interfaces. However, notice that these reflect the case where the medium is replaced by air, as described in Section 2.1.

**Figure 8 sensors-23-00385-f008:**
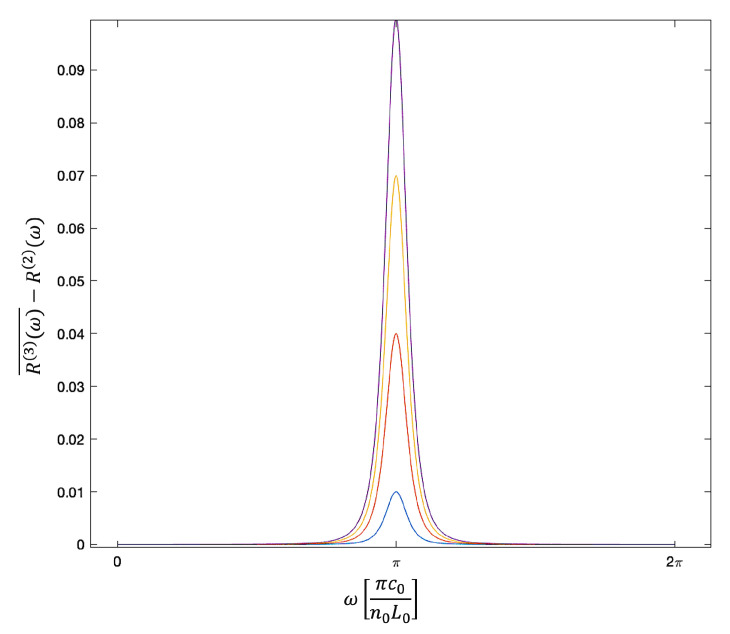
To illustrate the effect of randomly-positioned mirrors on the overall reflection rate of a Fabry-Pérot cavity, the figure shows the dependence of $\bar{R^{\left(3\right)}\left(\omega \right)}-R^{\left(2\right)}\left(\omega \right)$ in Equation (Equation 32) on the frequency ω of the incoming light. Here, $|r_{a}^{\left(2\right)}{|}^{2}={\left|r_{c}^{\left(2\right)}\right|}^{2}=0.81$, while $|r_{c}^{\left(3\right)}{|}^{2}=0.01$ (blue), 0.04 (red), 0.07 (yellow) and 0.1 (purple). The difference is a small and narrow upwards peak of a certain *full width at half the maximum* (FWHM) and with an amplitude given by $|r_{c}^{\left(3\right)}{|}^{2}$.

**Figure 9 sensors-23-00385-f009:**
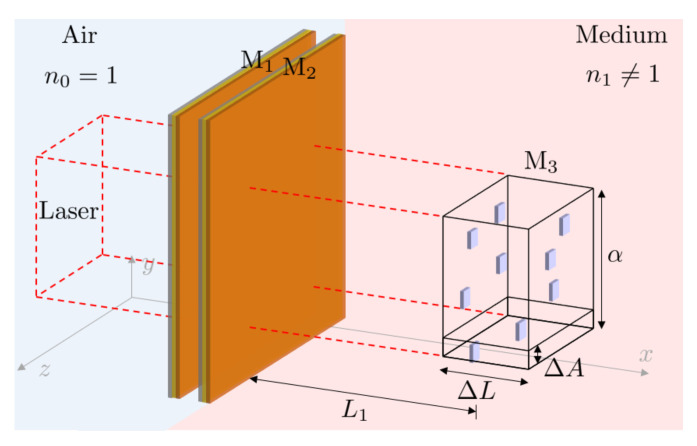
Schematic view of an experimental setup, which contains a Fabry-Pérot cavity as well as a group of tiny mirrors, which are randomly positioned in a medium with refractive index n≠1. These additional mirrors occupy a volume of length ΔL and covering an area α, which is placed some distance $L_{1}$ away from the cavity. An incoming laser with its cross section given by α approaches the cavity and the additional mirrors from a perpendicular direction. For simplicity, we assume here that the additional mirrors only cover a relatively small percentage of the area such that each one of them is likely to be observed by the incoming laser light.

**Figure 10 sensors-23-00385-f010:**
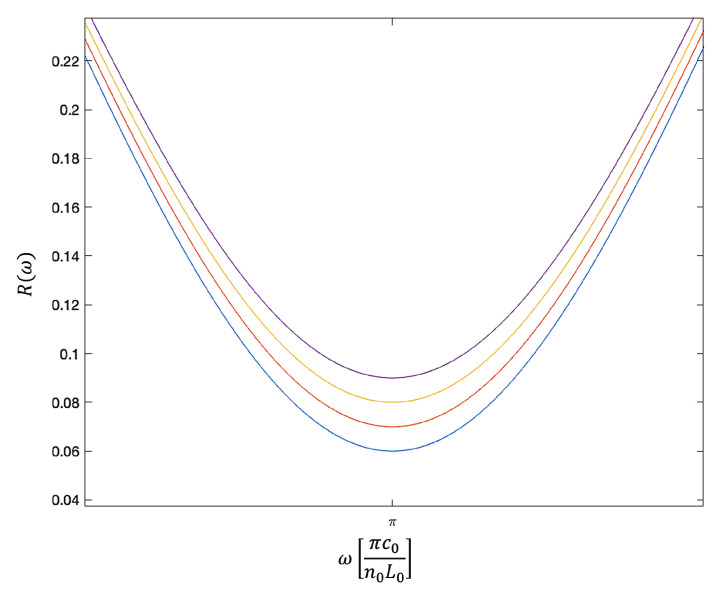
The dependence of the overall reflection rate R(ω) in Equation (Equation 37) of the Fabry-Pérot cavity in Figure 9 on the frequency ω of the incoming light. Here the mirror parameters are $|r_{a}^{\left(2\right)}{|}^{2}={\left|r_{c}^{\left(2\right)}\right|}^{2}=0.81$, while $|r_{c}^{\left(3\right)}{|}^{2}=0.1$. In addition, $P_{N}\left(\geq 1\right)$ equals 0.6 (blue), 0.7 (red), 0.8 (yellow) and 0.9 (purple). The graphs have again been calculated using Equations (Equation 23) and (Equation 32) and shows that the presence of a relatively large number of tiny, randomly-positioned mirrors increases the minimum of the reflection rate R(ω) of the Fabry-Pérot cavity. Instead of zero, the minimum now equals $R_{\min}$ in Equation (Equation 38).

**Figure 11 sensors-23-00385-f011:**
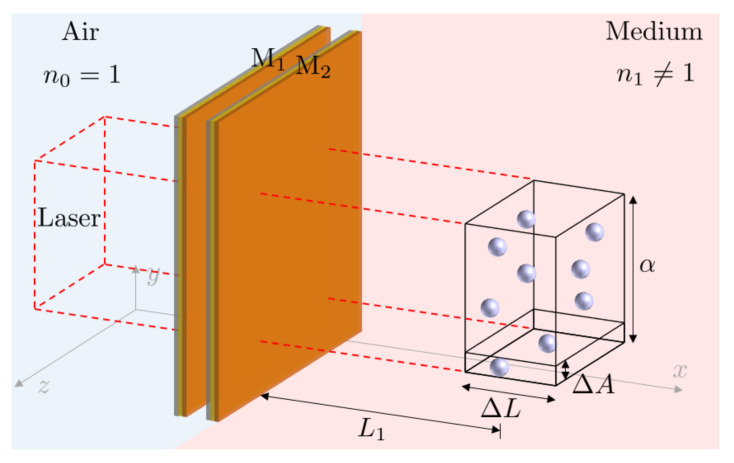
An alternative view on the remote Fabry-Pérot cavity sensor in Figure 2. Here, the target molecules are randomly distributed within a volume *V* of length ΔL a distance $L_{1}$ away from the resonator. Moreover, ΔA and α denote the cross section of a single molecule and α is the area that the incoming laser light excites. Our hypothesis here is that the molecules closely resemble tiny semitransparent mirrors, which suggest the same optical response of the above experimental setup and the experimental setup shown in Figure 9.

## Data Availability

Statement of compliance with EPSRC policy framework on research data: This publication is theoretical work that does not require supporting research data.

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
