# Peer review of "Remote Non-Invasive Fabry-Pérot Cavity Spectroscopy for Label-Free Sensing"

_sensors, 2022, doi:10.3390/s23010385_

Round 1
Reviewer 1 Report
his manuscript presents a theoretical consideration of the Fabry-Perot cavity sensor where (in contrast to traditional situations) the authors consider the substance some distance away from the resonator. The manuscript demands essential improvement.
Comments:
1. In the caption for Fig. 1, the words extrinsic and intrinsic should be interchanged
2. Eq. (2) does not depend on the direction of the light propagation. It should be the sign plus in the right hand part of the equation but not plus/minus
3. The n in Eq. (5) is called usually as the refraction index but not as "diffractive".
4. In the beginning of the paper it is not explained why the authors consider the substance some distance away from the resonator. This is explained only in the conclusions section.
5. Eq. (6) is not trivial. It should be derived or explained with more details.
6. The authors should repeatedly check the text of the manuscript to eliminate other possible errors
Author Response
We would like to thank the reviewer for their constructive comments. The whole manuscript has been carefully checked and re-written. The paper is now much easier-to-read and more motivation has been added.
- Thanks for pointing this out. The names of the sensor types have now been changed everywhere and a citation has been added to a textbook on the topic.
- This equation has been removed.
- This typo has been corrected.
- More details have now been added to the Introduction of the manuscript. Our main aim is to obtain a remote optical sensor which can be used to non-invasively and continuously monitor physical, biological and chemical processes. All we require is that the target molecules have optical transitions which reflect the incoming laser light while their environment is transparent—or at least semi-transparent—in the respective frequency range. The discussion of the potential attractive features of the proposed remote Fabry-Perot cavity sensor has been extended.
- More details have been added to explain the equivalence relation in Eq. (6) [which is now Eq. (5)].
- The whole manuscript has been carefully checked, improved and rewritten.
Reviewer 2 Report
The manuscript presents a remote non-invasive optical sensor for the label-free detection of bio-molecule concentrations.
Comments:
(1) The topic is fair bat I feel the manuscript needs to be reorganized. Some basic descriptions are presented in Sec. 4 which should be moved to another section.
(2) The figures should be presented in order. In the present form Figure 7 is presented after Figures 8 and 9.
(3) The conclusion should be concise and engaging. In the present form, the authors mentioned some figures and sections which needs to be revised.
(4) Label-free and label-based photonic biosensors are widely presented and therefore it is expected to use the more recent works to complete the literature survey, e.g. 10.3390/s21010089, 10.1016/j.optlastec.2021.107397 and 10.3390/s18103519.
(5) Please add some legends for the curves (for example: Figs. 6 and 8).
(6) Important parameters of biosensors should be presented and analyzed. For this reason, important parameters including quality factor, detection limit, sensitivity, figure of merit and so on are recommended to be considered.
Author Response
We would like to thank the reviewer for their very helpful comments. The manuscript has been very carefully re-organised and re-written. The comments of the referee have helped us to improve the paper and have all been taken into account.
(1) We agree with the referee and changed the structure of the manuscript. Section 2 now describes the optical response of a Fabry-Perot cavity which is the expected optical response of the proposed sensor in the absence of target molecules. Section 3, now analyses the effect of additional mirror and contains calculations which were previously included in Section 4. The discussion of the proposed remote sensor in Section 4 has been extended and only a minimum number of equations is presented here.
(2) The Figures are now in the correct order. Fig. 6 illustrates the results obtained in Section 2, while Fig. 8 illustrates the results obtained in Section 3. This figure now contains the numerical as well as the analytical results.
(3) The conclusions have been rewritten together with the rest of the paper.
(4) The references have been updated and the more recent reviews suggested by the referee have been included in the manuscript.
(5) Table 1 is the legend for Figs. 6 and 8. To make this table more visible, it now occurs before the figures in the text.
(6) The text now contains a much more detailed discussion of the relevant parameter regimes and the assumptions that have been made.
Reviewer 3 Report
The article described a so-called intrinic FP detection method. Detailed analysis was presented to explain the transmission and the reflection characteristics of two-mirror FP and three mirror FP. Conclusion was made that the intrinic FP can be used to detect the concentration of biomolecules by monitoring the reflectivity change of the cavity at resonance frequencies.
However, the article lacks significant content. What is the advantage when comparing the reflectivity change of the biomolecules with FP free situation? There lacks quantity comparison.The simulation of R2(w)-R3(w) as shown in fig.7 and fig.8 did not reflect such information.
It is not sound to assume the positions of biomolecules do not affect the reflectivity.
I do not think the article is suitable for publication in its current state.
Author Response
The manuscript has been carefully rewritten in response to the comments of the reviewer.
Remote Fabry-Perot cavity sensors have several advantages compared to extrinsic and intrinsic sensors and might therefore find interesting applications in the label-free and non-invasive detection of biomolecule concentrations. For example, remote sensors can be used to continuously and non-invasively monitor a wide range of physical, chemical and biological processes. No direct contact with the target molecules is required as long as they are optically accessible.
Another important advantage of remote Fabry-Perot cavity sensors is that they do not need to be adjustable, since they do not need to be stabilised in direct contact with target molecules. This means, they can be fabricated more easily, for example by integrating them into optical fibres, while accompanying them by integrated mode-matching optics.
It is difficult to present a quantitative comparison of the different sensors, since they operate in different ways. Traditional Fabry-Perot cavity sensors deduce information by comparing frequency shifts and the broadening of the line width of the transmitted light to a known baseline. In contrast to this, remote Fabry-Perot cavities measure biomolecule concentrations by observing a small upwards peak that is added to the minimum of their reflection spectrum. The size of this upwards peak provides information about molecular concentrations.
Also, the referee is right and the reflectivity of the sensor depends in general on the positions of the target molecules. However, here we assume that the positions of the target molecules within the sample are completely random. In this way, all reflected light accumulates a random phase. The reflection rate R(omega) of the sensor is obtained by averaging over these random phases. As a result, the reflectivity in Eq. (32) depends only on the concentration C of the biomolecules, their scattering cross section ∆A and the optical depth ∆L of the sample.
Overall, we feel that the manuscript has been significantly improved and we would like to ask the referee to re-consider our paper for publication.
Round 2
Reviewer 2 Report
The revised manuscript can be accepted for publication in Biosensors.
Author Response
Reviewer 2 already recommended publication of our manuscript. However, they also mentioned that some aspects of the manuscript can be improved. In response, we carefully rewrote parts of the Introduction of the manuscript. Most importantly, we improved the calculations in Sections 2 and 3 and are now able to present an exact analytical expression for the reflection rate of a Fabry-Perot cavity sensor in the presence of a large collection of randomly-positioned tiny mirrors on one side of the resonator (cf. Eq. (31)). We also extended the discussion of the potential use of remote Fabry-Perot cavity sensors in non-invasive sensing in Section 4 and better illustrate the effects of the target molecules in an additional figure (cf. Fig. 10). Overall, the writing of the manuscript has been improved.
Reviewer 3 Report
This is a pure theoretical article. It can be accepted in its present form. It would be better if expermental results could be compared.
Author Response
In their report, the reviewer mentioned that the results could be presented more clearly and that our conclusions could be supported better by our results. In response to the reviewer, as in our response to Reviewer 2, we improved the calculations in Sections 2 and 3 and are now able to present an exact analytical expression for the reflection rate of a Fabry-Perot cavity sensor in the presence of a large collection of randomly-positioned tiny mirrors on one side of the resonator (cf. Eq. (31)). We now also illustrate the effects of the target molecules on the measurement signal in an additional figure (cf. Fig. 10). Overall, the writing of the manuscript has been improved.